# Structural Characteristics and Immunomodulatory Effects of Melanoidins from Black Garlic

**DOI:** 10.3390/foods12102004

**Published:** 2023-05-15

**Authors:** Xiwang Song, Liangyu Xue, Xiaoyuan Geng, Jianfu Wu, Tao Wu, Min Zhang

**Affiliations:** 1State Key Laboratory of Food Nutrition and Safety, Food Biotechnology Engineering Research Center of Ministry of Education, College of Food Science and Engineering, Tianjin University of Science & Technology, Tianjin 300457, China; s9249527@163.com (X.S.);; 2China-Russia Agricultural Products Processing Joint Laboratory, Tianjin Agricultural University, Tianjin 300384, China

**Keywords:** black garlic, melanoidins, structural properties, immunity enhancers

## Abstract

Melanoidins are considered to have several biological activities. In this study, black garlic melanoidins (MLDs) were collected using ethanol solution extraction; 0%, 20%, and 40% ethanol solutions were used for chromatography. Three kinds of melanoidins were produced by macroporous resin, named MLD-0, MLD-20, and MLD-40. The molecular weight was determined, and the infrared and microscopic structures were studied. In addition, Balb/c mice were induced with cyclophosphamide (CTX) to establish an immune deficiency model to evaluate the immune efficacy of black garlic melanoidins (MLDs). The results showed that MLDs restored the proliferation and phagocytosis ability of macrophages, and the proliferation activity of B lymphocytes in the MD group was 63.32% (♀) and 58.11% (♂) higher than that in the CTX group, respectively. In addition, MLDs alleviated the abnormal expression of serum factors such as IFN-γ, IL-10, and TNF-α. 16SrDNA sequencing of intestinal fecal samples of mice showed that MLDs changed the structure and quantity of intestinal flora, and especially that the relative abundance of Bacteroidaceae was significantly increased. The relative abundance of Staphylococcaceae was significantly reduced. These results showed that MLDs improved the diversity of intestinal flora in mice, and improved the adverse state of immune organs and immune cells. The experiments confirm that black garlic melanoidins have potential value in immune activity, which provides an important basis for the development and utilization of melioidosis.

## 1. Introduction

The immune system is the rigorous defense mechanism of a host that resists antigen invasion and maintains body balance [1]. In addition to bacteria [2] and viruses [3], physiological and emotional fluctuations will cause disorders of the human immune system [4], and it is particularly important to maintain the homeostasis of the immune system. How to improve the immune ability of self-regulation has been gaining more and more attention. Natural ingredients have always been favored [5]. Improving gut microbiota through personalized nutrition and supplements as a preventive method can improve immunity, which may be a good way to help protect immunocompromised patients [6].

Black garlic is the self-fermented product of garlic in high-temperature and humidity environments [7]. In recent years, black garlic has been a hot topic and is recognized as a healthy food [8]. During the processing of black garlic, a new substance, black garlic melanoidins, was produced along with complex physicochemical changes. It gives the food its distinctive brown appearance and also adds an attractive flavor.

Black garlic is thought to have stronger antioxidant properties than garlic [9] and has a variety of effects: anti-inflammatory [10], reduces blood pressure [11], anti-fatigue [12], regulates blood sugar and lipid [13], inhibits obesity [14], etc. In addition, it was found that the extract of black garlic had immune-enhancing ability [15] and anti-tumor ability [16]. As the main component of the extract, the important contribution of melanoidins in enhancing immune activity was indirectly confirmed. The appearance of melanoidins is one of the important factors in the physiological activity of black garlic, which gives black garlic a higher edible value than that of garlic. At present, most reports are widely focused on the antioxidant, antibacterial, and anti-inflammatory effects of melanoidins, and the effects of melanoidins on immune function are rarely reported locally and abroad.

In this study, the melanoidins were extracted by ethanol solution, and the related physical and chemical properties of MLDs were examined by size exclusion chromatography, Fourier transform infrared spectroscopy, scanning electron microscopy, DSC, and other techniques. The purpose of the animal experimental study was to explore the immune activity of melanoidins from black garlic and provide a reference for the development and utilization of melanoidins in food, medicine, and other fields.

## 2. Materials and Methods

### 2.1. Materials and Reagents

Black garlic was provided by Laiwu Yuyuan Food Co., LTD. (Laiwu, China). Dialysis membrane (MWCO: 1000 Da) was purchased from Scientific Research Special Company (Atlanta, GA, USA). DA201-C macroporous resin was purchased from Shanghai Yuanye Biotechnology Co., LTD. (Shanghai, China) RPMI 1640, fetal bovine serum (FBS), penicillin, and streptomycin were provided by Hyclone; phosphate-buffered saline (PBS, pH 7.4), 3-(4,5-dimethylthiazol-2-yl)-2,5-diphenyltetrazolium bromide MTT, Concanavalin A (ConA), and Lipopolysaccharide (LPS) were provided by Sigma-Aldrich (St. Louis, MO, USA). All the other chemicals and reagents used in the experiment were analytical grade or highest purity.

### 2.2. Response Surface Experiment on Extraction of Melanoidins from Black Garlic

At present, the structure of melanoidins is unknown, and the water solution method is often used in the extraction and preparation of melanoidins. Four conditions, such as reaction temperature, extraction time, solid–liquid ratio, and ethanol solution concentration, had significant effects on the extraction of melanoidins. The absorption value was detected at UV 420 nm, and the extraction amount AV value was used as the determination of the extraction amount of melanoidins [17]. The extraction parameters are shown in Table 1.

### 2.3. Extraction of Melanoidins Samples

After peeling the same batch of black garlic, the melanoidins were extracted and lyophilized according to the optimal extraction conditions obtained from the response surface. The black garlic and ethanol solution were placed in a blender (TDL-5-A, Joyang, Jinan, China) in various proportions. After being evenly broken, the turbid solution was heated under the best conditions to extract melanoidins. At the end of the process, it was centrifuged at 3500 r/min for 15 min (TDL-5-A, Xiangyi, Changsha, China), and the precipitate was taken and processed again according to the above steps. The supernatant without precipitate was collected two times, filtered by filter paper, and combined. The samples were then dialyzed in tap water for 48 h and in distilled water for another 24 h, during which the distilled water was changed twice. Small molecules such as reducing sugar were removed after dialysis. A rotary evaporator (RE-52AA, Yarong, Shanghai, China) was used to vaporize samples at 40 degrees, and the samples with more than 1 kDa were concentrated and lyophilized.

The content of melanotype in this method is about 86%, and there are no other active ingredients.

### 2.4. Purification of Melanoidins Samples

DA201-C macroporous resin (Yuanye, China) was filled into the chromatographic column (Ø2.6 × 30 cm). The newly prepared melanoidin sample was dissolved in distilled water and prepared in a solution of 20 mg/mL. An amount of 20 mL was taken for wet sampling (the flow rate was 1.25 mL). The eluent was collected and concentrated, and the solvent was removed and freeze-dried for use in subsequent experiments.

### 2.5. Determination of the Molecular Weight of Melanoidin Samples

The method of Zhang [18] was referred to and modified, and the molecular weight of melanoidins was estimated by size-exclusion high-performance liquid chromatography. Protein dispersions were prepared in 50 mmol/L sodium phosphate buffer (pH 7.2) containing 0.5% wt% sodium dodecyl sulfate and mixed thoroughly. Then, 2 mg/mL of melanoidin samples was filtered through a 0.22 μm filter membrane and injected into the system, eluted at 0.6 mL/min, and the elution curve of the sample at 420 nm was recorded. The standards γ-globulin (158 kDa), bovine serum albumin (66 kDa), ovalbumin (44,287 Da), myoglobin (16,951 Da), and vitamin B12 (1355.37 Da) were used to obtain the molecular weight calibration curve.

Shimadzu LC-20 high-performance liquid chromatography system (Shimazu Scientific Instruments, Kyoto, Japan) was used for the experiment, equipped with a UV–vis detector and TSK Gel 6000 gel column (300 × 7.8 mm^2^; Tosoh Co., Tokyo, Japan), and the column temperature was 30 °C.

### 2.6. Infrared Analysis

Fourier infrared spectroscopy (Thermo Nicolet, Waltham, MA, USA) was used to measure the absorption spectra of melanoidins in the wavenumber range of 400–4000 cm^−1^. The dried melanoidin powder sample of 1.0 ± 0.1 mg was added into the mortar, followed by the weighing of 120 mg KBr, fully ground, mixed, and evenly transferred into the mold and squeezed into a piece. The absorption spectra of the three kinds of melanoidins samples were determined.

### 2.7. Microstructure Observation

The MLD was evenly coated on the sample table and the surface was sprayed with thin gold. Scanning electron microscopy (SU1510, Hitachi, Tokyo, Japan) was used to observe the morphological distribution. The images were taken at 1000 times magnification.

### 2.8. DSC Analysis

DSC (TA Instruments, New Castle, DE, USA) was equipped with a TA-60 WS acquisition and monitoring system (Shimazu Scientific Instruments, Kyoto, Japan). A total of 2 mg of melanoidins was taken and buried in an aluminum crucible. The experimental temperature was set at 30–550 °C and the heating rate was set at 10 °C/min. The denatured peak temperature was analyzed by differential scanning calorimetry.

### 2.9. Design of Experimental Animals

SPF Balb/c mice (♀:18 ± 2 g, ♂:22 ± 2 g) at 6 weeks of age were used in this study, with 40 females and 40 males provided by Sibeford Biotechnology Co., LTD. (Beijing, China). The room temperature of 23 ± 1 °C, and a 12 h light/dark cycle were set. The experiment was conducted in strict accordance with the guidelines of the Animal Experiment Ethics Committee of Tianjin University of Science and Technology (TUST20200108).

Before the experiment, all the mice were fed adaptively for one week, with free access to food and water, and then randomly divided into five groups with 16 mice in each group, 8 males and 8 females. The groups were set as follows: normal group (NC), model group (CTX), low-dose MLD group (LD), medium-dose MLD group (MD), and high-dose MLD group (HD). The experiment lasted for 16 days. The mice in the three dose groups were intragastrically given 0.2 mL melanoidin solution (50, 100, 200 mg/kg·bw), and the other two groups were intragastrically given 0.2 mL normal saline. On the 8th day of the experiment, all mice except the normal control group were intraperitoneally injected with 80 mg/kg.bw cyclophosphamide saline solution for three consecutive days [19]. After the last intragastric administration, feeding was stopped, and sacrificing was performed 12 h later.

### 2.10. Measurement of Organ Index

The mice were weighed at the beginning and at the end of the experiment. The whole spleen and thymus of the mice were removed surgically (after the blood stains were washed with normal saline, the water was dried with filter paper) and weighed and recorded. The organ index of the mice was calculated according to the following formula [20]:Index (mg/g) = immune organ weight (mg)/body weight (g)

### 2.11. H&E Staining of Organs

Standard techniques were used for H&E staining. The spleen of the mice was extracted, and the excess fat and interfering tissue were removed and fixed in 4% paraformaldehyde. After paraffin embedding, dewaxing section, and H&E staining, the cell morphology was observed and photographed using an inverted fluorescence microscope (CKX41, Olympus, Tokyo, Japan).

### 2.12. Determination of the Activity of Mouse Peritoneal Macrophages

Under sterile conditions, 5 mL PBS buffer was injected into the intraperitoneal cavity of mice, ascites were absorbed, and precipitated macrophages were collected by centrifugation. The cell density was adjusted to 2 × 10^6^ cells /mL by adding RPMI-1640 containing 10% thermal inactivated FBS and 1% penicillin–streptomycin.

#### 2.12.1. Determination of the Proliferative Activity of Macrophages

An amount of 100 μL macrophages (2 × 10^6^ cells /mL) was inserted into each well of the 96-well plate and cultured for 48 h at 37 °C under 5% CO_2_ (HF240, Lixin, China); then, 2 µL MTT solution (5 mg/mL) was added to each well, and 150 µL DMSO was added to each well after 4 h of culture. The absorbance value at 570 nm was determined by a microplate reader (Thermo, Multiskan, FC, USA) [21].

#### 2.12.2. Determination of the Phagocytosis Ability of Macrophages

An amount of 10 µL peritoneal macrophages (2 × 10^6^ cells /mL) was inserted into each 96-well plate, and RPMI-1640 complete medium was used as blank control. The cells were cultured for 3 h in a cell incubator (HF240, Lixin, China) under the conditions of 37 °C and 5% CO_2_, and then the 96-well plates were washed twice with PBS buffer solution to exclude nonadherent cells. After that, 100 µL 0.1% (*m*/*v*) neutral red solution was added to each well, incubated for 3 h, and washed three times with pre-warmed PBS to remove the remaining neutral red. After treatment, 100 µL cell lysate (anhydrous ethanol:acetic acid =1:1, *v*/*v*) was added to each well, and the absorbance at 540 nm was measured using an enzyme label (Thermo, Multiskan, FC, USA) [22].

### 2.13. Determination of the Activity of Mouse Spleen Lymphocytes

The spleen is the body’s largest immune organ. It is the site where lymphocytes mature and act. After sacrificing mice, the spleen was removed under aseptic conditions, the cells were dispersed by extrusion, the red blood cells were cleaved and removed, and then transferred to a 37 °C, 5% CO_2_ incubator for 2 h culture (HF240, Lixin, China). The nonadherent spleen lymphocytes were purified [23].

#### 2.13.1. Proliferative Activity of Mouse Spleen Lymphocytes

Purified spleen cells were added with RPMI-1640 and adjusted to a certain cell concentration (5 × 10^7^ cells /mL) in complete medium, and 100 µL was added to each hole on the 96-well plate. T lymphocytes and B lymphocytes were induced by ConA (5 μg/mL) and LPS (5 μg/mL), respectively [24]. The cells were cultured in a cell incubator (HF240, Lixin, China) at 37 °C under 5% CO_2_ conditions for 48 h; 20 µL MTT solution (5 mg/mL) was added to each well [25], the supernatant was discarded after continued culture for 4 h, and 150 µL DMSO was added to each well. After full shock, the absorbance was measured at 570 nm using an enzyme marker (Thermo, Multiskan, FC, USA). The proliferation capacity of lymphocytes was determined by subtracting the absorbance values of the blank control group from the measured absorbance values of the T lymphocyte group or the B lymphocyte group [26].

#### 2.13.2. Determination of the NK Cell-Killing Activity of Spleen Lymphocytes in Mice

In this study, the LDH method was used to measure the activity of NK cells, and some modifications were made [27]. In brief, the experimental response group was set up with S180 mouse tumor cells as target cells (1 × 10^4^ cells/mL) and spleen cells as effector cells (1 × 10^6^ cells /mL), and the effect target ratio was 100:1. At the same time, a natural release group and maximum release group were set to detect the killing activity of NK cells [28]; each had 6 parallel multiple holes.

The NK response group was supplemented with 100 µL target cells and 100 µL spleen cells. The natural release group was supplemented with 100 µL target cells and 100 µL RPMI-1640 complete medium. In the maximum release group, 100 µL target cells and 100 µL 1% NP40 lysate were added.

The 96-well plates of each group were placed in an incubator (HF240, Lixin, China) under the conditions of 37 °C and 5% CO_2_ for 48 h, and then the whole plate was centrifuged (TD3, Xiangyi, China) for 5 min (1000 r/min). The supernatant and LDH matrix solution were mixed with 100 µL each at a volume ratio of 1:1, and after standing for 5 min for a full reaction, 30 µL HCl (1 mol/L) was added to each well. A horizontal vibrator was used for shock for 5 min, and the absorbance value at 490 nm was measured with an enzyme label [29].

The final NK cell-killing activity was calculated by the following formula:NK activity (%) = (ODexperimental − ODspontaneous)/(ODmaximum − ODspontaneous)

### 2.14. Determination of Related Immune Factors in the Serum of Mice

Mouse orbital blood was taken after centrifugation at 1500 rmp for 10 min (Micro 17R, Thermo, Dreieich, Germany), and serum samples were collected and stored at −80 °C until analysis. The contents of IL-10, TNF-α, and IFN-γ in serum were determined by an ELISA kit, following the instructions to calculate the expression of the corresponding index in the sample.

### 2.15. Determination of SCFA in the Mouse Intestinal Short-Chain Fatty Acids

The measurement of SCFAs referred to the method of Zhu [30] and made appropriate modifications. Gas chromatography conditions are as follows: Agilent 7890A GC system; DB-WAX column (Agilent 19091N-133; 230 °C; 30 m × 250 mm × 0.25 μm; in: back SS inlet N2; out: back detector FID) and a flame ionization detector (Agilent Technologies Inc., Santa Clara, CA, USA). The acetic acid, propionic acid, butyric acid, and valerate standard products were mixed with ether and prepared into a mixed standard solution of 10 μL/mL SCFAs, diluted with ether (1, 0.1, 0.2, 0.3, 0.4, and 0.5 µL/mL), and then injected successively. Standard curves were drawn according to the concentration as the abscissa (X) and the standard peak area as the ordinate (Y).

After the mice were sacrificed, the cecum contents were obtained, lyophilized, weighed 0.05 g, mixed with 0.5 mL saturated NaCl solution, and then homogenized after standing at room temperature for 30 min. After acidizing with 40 µL concentrated sulfuric acid, it was mixed in a vortex oscillator for 30 s; then 0.8 mL ether was added to extract short-chain fatty acids, and the supernatant was taken for 30 s, centrifuged (12,000× *g*, 4 °C, 15 min), and added to a centrifuge tube containing 0.25 g anhydrous NaSO_4_ for 10 min to absorb residual moisture. After centrifugation, the supernatant was transferred to a gas phase vial for analysis by gas chromatography and mass spectrometry (GC-MS).

### 2.16. Determination of Intestinal Flora in Mice

After the mice were killed, the colon contents of the mice were taken into a sterile EP tube. The V3-V4 variable region of the 16S rRNA gene was amplified by PCR. OTU clustering and species classification analysis were conducted based on the valid data after processing the original data.

### 2.17. Data Analysis

SPSS 24 software (IBM SPSS Statistics 24.0) was used for data analysis and processing, and the experimental results were expressed as mean ± standard deviation. The data results between groups were analyzed by one-way analysis of variance (ANOVA), and *p* < 0.05 was considered a significant difference; *p* < 0.01 was considered a significant difference.

## 3. Results and Discussion

### 3.1. Extraction and Optimization of MLDs

The response surface results extracted by MLDs are shown in (Figure 1). After several numerical optimizations, the predicted optimal conditions are as follows: a heating time of 54 min, a heating temperature of 46 degrees, a solid–liquid ratio of 1:17.6, and an ethanol solution concentration of 14.2%. The validity of the model was verified by experiments.

### 3.2. Separation and Purification of Melanoidins

Black garlic melanoidins were purified by DA201-C macroporous resin, and the solutions in tubes at the elution peak were collected and named MLD-0, MLD-20, and MLD-40, in turn. The elution curve is shown in Figure 2.

### 3.3. Melanoidin Molecular Weight

The molecular weight standard curve y = −0.3492x + 9.84519 (R2 = 0.995) was calculated by using standard materials, where y was the logarithm of molecular weight and x was the retention time. The molecular weight of black-like concentrate was calculated according to the retention curve, as shown in Table 2.

### 3.4. MLD Infrared Analysis

The infrared absorption spectrum, which can analyze the composition of substances, is often used to identify the structure of organic matter. By analyzing the characteristic absorption frequency of important functional groups, the structure of the compounds was identified, and qualitative and quantitative analysis was carried out.

It can be observed from Figure 3 that the melanoidin infrared spectra obtained by elution at different concentrations are generally similar. The appearance of the absorption band at 3400 cm^−1^ is due to O-H stretching. At 2927 cm^−1^, the peak is -CH or -CH2 tensile vibration. The absorption band of the whole band can confirm that the melanotype contains different material compositions, being a complex mixture of components. The weak absorption band near 1612 cm^−1^ may be caused by C=O stretching vibration, N-H bending vibration, or C-N stretching vibration [31], it may be caused by C-N deformation at 1400 cm^−1^.

The absorption peaks of C-O tensile vibration, **C**-C tensile vibration, and C-H bending vibration may appear at 1050 cm^−1^. Zhang [32] observed a melanoidin extract from dark beer with infrared and obtained similar conclusions. This is most likely due to the consistency of the MLDs structural framework, even though the raw materials used to produce melanoid are different, and the difference exists in the fingerprint region below 1330 cm^−1^.

### 3.5. SEM Observation

The morphological characteristics of different components were determined by scanning electron microscopy, and the results are shown in Figure 4. An irregularly curling pattern was observed for the 0% elution, an irregular fragmentary pattern for the 20% elution, and a flat sheet pattern for the 40% elution. The three elution compositions are obviously different.

### 3.6. Thermal Analysis of MLD

The results of DSC are shown in Figure 5. The components MLD-0, MLD-20, and MLD-40 were collected by using distilled water, 20%, and 40% ethanol solution as eluent, in turn. According to the DSC pyrolysis curves of each component, the pyrolysis peak of component MLD-0 at 300~400 °C can be attributed to the decomposition of the structure of cellulosic polysaccharide, indicating that MLD-0 is mainly composed of hydrophilic substances, with the main chain of sugar rings or oligosaccharides generated during the Maillard reaction. Two distinct pyrolysis peaks were observed in the range of 230–340 °C and 350–530 °C for both MLD-20 and MLD-40, which could be attributed to the oxidation of methyl and the degradation of sugar ring opening, and the oxidation of side-chain carboxyl, respectively. The enthalpy change values of MLD-20 were 743.71 J/g and 19.72 kJ/g, respectively. The enthalpy changes of MLD-40 were 2.7 kJ/g and 16.74 kJ/g, respectively. Compared with MLD-0, MLD-20 and MLD-40 have higher thermal stability. The results showed that MLD-40 was mainly composed of weak hydrophilic substances with a carboxyl group and methoxy group in the side chain and a sugar ring group in the main chain. MLD-20 is mainly composed of hydrophobic monosaccharide rings or oligosaccharides with a small amount of carboxyl group and a large amount of methoxy group as side chains.

### 3.7. Effects of MLDs on Body Weight and Organ Index in Mice

CTX is the most commonly used immunosuppressant [33] for modeling immunodeficient mice. The body weight of experimental mice is shown in Table 3. The body weight of male and female mice in the CTX group was significantly lower than that in the NC group (*p* < 0.01). As can be seen in the table, the body weight of mice in each dose group was increased compared with that in the CTX group. LD and MD had significant effects on weight recovery, and the weight of male and female mice in the MD group was 8.66% (female, *p* < 0.05) and 14.92% (male, *p* < 0.01) higher than that in the CTX group. It was proved that MLD had a better effect on weight recovery in immunosuppressed mice.

The spleen and thymus are important immune organs of the body, and the index of the spleen and thymus provides a rough estimate of non-specific immunity, which can reflect the immune function of the body to a certain extent [34]. The spleen index and thymus index are also shown in Table 3. The spleen index of both male and female mice was higher than that of the CTX group and showed an increasing trend with the increase in dose. The indices of the high-dose group were all significant (*p* < 0.01). As for the thymus index, there was an increase in each dose group compared with the CTX group, but the result was not significant due to the small thymus and the possibility of incomplete loss or tissue removal.

Overall, MLDs have a good effect on body weight recovery and can alleviate organ damage caused by CTX. Compared with female mice, male mice showed greater flexibility in weight change in response to CTX induction and MLD feeding, which may be related to the metabolic strength of the body and may also be caused by differences in gender opposition. Pennell et al. [35] have shown that sex differences affect innate and adaptive immune responses, not only because of sex hormones but also because of genes and environment.

### 3.8. Effect of MLDs on Spleen Morphology in Immunocompromised Mice

The effects of MLDs on immune organs were evaluated by mouse spleen slices, and the results are shown in Figure 6. It was observed that the splenic sections of mice in the NC group showed regular cell morphology and better integrity of each region. In the CTX group, more lymph nodules could be observed, but the structure of the white pulp area was chaotic, and the boundary between the white pulp and the red pulp as well as the marginal area was blurred. The cell gap was large, the cell size was uneven, and the morphological difference was significant. The LD group also had more lymph nodules, but the cell morphology and structure were significantly relieved and improved compared with the CTX group. In the MD group, the white and red pulp regions were clearly visible, the boundary of the marginal regions was clear, the tissue structure was complete, and the cells were closely arranged and in good condition. Compared with the CTX group, the cell morphology of the HD group was also improved. Histopathological analysis indicated that the melanoidins of black garlic had better recovery ability than the spleen.

### 3.9. The Effect of Black Garlic Melanoidins on the Abdominal Macrophage in Mice

#### 3.9.1. The Effect of MLDs on the Proliferation of Macrophages in Mice

The macrophage is an important part of the body’s immune regulation, playing a role in specific immunity and non-specific immunity. Phagocytosis is the main method of eliminating the alien and mediate non-specific immunity. It is also an integral part of specific immunity, mediating specific immunity by presenting antigenic substances and exposing antigenic epitopes. Meanwhile, its proliferative activity and phagocytic ability are important indicators to measure its immune activity. MTT colorimetry was used to determine the viability of living cells by measuring the activity of mitochondrial dehydrogenase. The effect of MLDs on the proliferative activity of mouse macrophages is shown in Figure 7.

Observation of all groups showed that induction of CTX significantly decreased the proliferation capacity of mouse macrophages, and the proliferation capacity of mouse macrophages was better improved after feeding with black garlic melanoidins. Compared with the model group, the therapeutic effect on all mice in the medium-dose and high-dose groups was extremely significant, and the medium-dose group had the best effect. Therefore, it can be seen that MLDs have a good effect in restoring the proliferative activity of mouse abdominal macrophages induced by CTX. The overall accretion capacity measured in males was significantly stronger than that of females.

#### 3.9.2. Effect of MLDs on the Phagocytic Ability of Mouse Peritoneal Macrophages

The effect of MLDs on the phagocytosis capacity of mice peritoneal macrophages is shown in Figure 8. Compared with the CTX group, the phagocytosis capacity of mice in the dose group was significantly increased (*p* < 0.05). The images showed that there was a high degree of agreement between males and females in each group, and the differences were not obvious. It was also noted that male mice were more sensitive to CTX than female mice and had a higher phagocytosis recovery space in MD. It is speculated that female mice have higher resistance to CTX, and MLDs have a better recovery effect on immunodeficient male mice. This is consistent with the previous analysis based on the weight data. It is widely believed that there are differences in the immune system between males and females, with males generally more susceptible to infection, possibly due to the fact that sex steroids can affect immune cell function by binding to specific receptors expressed on cells [36].

### 3.10. Effect of MLDs on Spleen Cells in Mice

#### 3.10.1. Effect of MLDs on the Proliferative Activity of Spleen Lymphocytes in Mice

The effect of MLDs on the proliferative activity of mouse spleen cells is shown in Figure 9A. All groups showed that female mice had higher B cell proliferation activity than male mice. The proliferative activity of mice in the three dose groups increased first and then decreased, the proliferative ability of mice in the MD group and the HD group was significantly increased, and the proliferative activity of B cells in the MD group was higher than that in CTX group by 63.32% (♀) and 58.11% (♂).

There was a significant difference in T cell proliferation activity between male and female mice (Figure 9B). The proliferative ability T cells of male in the NC group was higher than that in females. However, after induction of CTX, the proliferation activity of female T cells was higher than that of male cells in the CTX group and the three dose groups. At the same time, it was observed that the increment ability of females in the dose group decreased gradually with the increase in feeding dose, but that of males showed a trend of first increasing and then decreasing, as in B cells. However, they were all higher than the CTX group, which confirmed that MLDs had a better effect on the proliferation ability of T cells.

In terms of the proliferative activity of splenic lymphocytes, MLDs have a good effect on the proliferative activity of spleen B cells and T cells.

#### 3.10.2. Effect of MLDs on the NK Cell-Killing Activity in Mice

NK cells are an important part of the body’s immunity, which can recognize and kill tumor and virus-infected cells without prior sensitization, and coordinate and promote immune response [37,38]. NK cells secrete a variety of substances that promote immune efficacy and play a key role in eliminating harmful cells, such as tumor cells and virus-infected cells.

The experimental results of NK cell-killing activity in mice are shown in Figure 10. Compared with the CTX group, the activity of cells in the LD group increased, but there was no significant difference between male and female mice. The effect of the MD and HD groups was significantly higher than that of the CTX group (*p* < 0.01). Among them, the MD group had the best effect on restoring NK cell-killing activity, while the HD group showed a decrease. In conclusion, MLDs can promote the recovery of NK cell-killing activity in mice, and except for the MD group, the killing activity in male mice is higher than that of female mice. MLDs can restore the function of immune organs and improve the activity of immune cells to a certain extent. Kang et al. [39] found that supplementation of whey protein after the Maillard reaction significantly increased the activity level of NK cells in subjects.

### 3.11. Determination of Related Immune Factors in the Serum of Mice

The immune factor can be regarded as a response to the level of humoral immune strength. Serum levels of related immune factors in mice were measured by ELISA, as shown in Figure 11. After induction of CTX, the serum IL-10 level was significantly higher than that of normal mice. In addition, after feeding MLDs, the serum IL-10 level of mice in all dose groups decreased, except for the low-dose group, in which it was significantly lower than in the CTX group (*p* < 0.01). After induction of CTX, the levels of TNF-α and IFN-γ would remain at a low level. The presence of MLDs may modulate this condition by moderating the levels of related factors present in the serum after induction.

### 3.12. Determination of Short Chain Fatty Acids in Mouse Intestine

The changes in the health state and diet structure of the host body will affect the corresponding beneficial or harmful changes in the intestinal flora. Once affected by external factors, the dynamic balance between the intestinal flora and the host will be destroyed, resulting in flora disorder [40], mainly in its structure and quantity. At the same time, this will be accompanied by a variety of metabolic and immune diseases. Intestinal flora is a product of different metabolic patterns that may be involved in the regulation of the immune system. In in vitro culture, melanoidins increased the production of short-chain fatty acids in the microbiome and led to significant changes in microbial colony structure [41].

The contents of SCFAs are shown in Table 4. The induction of CTX affects the contents of four kinds of SCFAs. There was a significant difference in acetic acid content between the HD group and the CTX group, and no significant difference between other groups and the CTX group. After CTX induction, the propionic acid content in the intestinal tract of mice was proportional to the dose of MLD fed, and both of them recovered well. The content of valerate was similar to that of propionic acid, and the content of valerate in the three groups was positively correlated with the increase in the content of melanin. The content of butyric acid in the HD group was close to that in the NC group, and significantly higher than that in the CTX group (*p* < 0.01). The content of SCFAs in the CTX group was significantly lower than that in the other groups, indicating that the modeling was successful and different bacterial communities producing SCFAs were decomposed and changed, which could be further analyzed in combination with 16S.

### 3.13. Determination of Intestinal Flora in Mice

There are a large number and a wide variety of microbial flora in the intestinal tract of animals. The microbiota plays a key role in the proper functioning of the digestive tract and maintaining the homeostasis of the organism [42,43], and the gut microbiome is therefore often referred to as the “second human genome” [44]. With the in-depth study of intestinal flora, researchers have found that a variety of metabolic diseases are significantly associated with the status of intestinal flora. Bad living habits and related external environmental factors will promote the expansion of some intestinal bacteria that quickly occupy the dominant position. When the disordered state of intestinal flora is formed, the dominant species will give feedback signals to the host, thus promoting the expansion and stability of the flora, which shows a detailed correlation in relevant studies [45].

The microbiome plays a key role in the formation and development of host innate and adaptive immunity, and the immune system promotes the balance and maintenance of host–microbe symbiosis [44,46]. In addition to providing nutrients, diet also transmits antigens. Food intake and microbes are factors that determine whether the immune system is mature [47,48]. Cahenzli et al. [49] reported that the diversity of intestinal flora colonized in the early stage directly affected the formation of the immune regulatory network. With the rapid development of high-throughput sequencing technology, more and more studies have found that there are significant differences between the intestinal flora of patients with a variety of autoimmune diseases and healthy people, mainly in the reduction of bacterial abundance and the increase in specific strains. 16S rRNA is a subunit of ribosomal RNA. The 16S rDNA encoding this subunit has 10 conserved regions and 9 hypervariable regions. Because the latter are specific to different genera or species, 16S rDNA can be used as a characteristic nucleic acid sequence for the classification and identification of microorganisms [50].

According to OTU results and research requirements, the common and unique OTUs among different samples (groups) were analyzed, and a petal diagram was generated in which each circle represented a sample (group). The number of circles and overlapped parts represents the number of common OTUs among samples (groups), while the number of non-overlapped parts represents the number of unique OTUs among samples (groups). The results (Figure 12) showed that the number of OTUs decreased after cyclophosphamide modeling, and increased after MLD treatment, indicating that MLD treatment can change the number of intestinal microorganisms in mice.

The Shannon index is one of the indexes often used to represent microbial diversity. The larger the value, the higher the microbial diversity. As can be seen from the results in Figure 13A, after CTX stimulation, the Shannon index in the model group decreased significantly, indicating that the microbial diversity in the intestinal tract of animals in the CTX group decreased after modeling. Compared with the CTX group, the Shannon indices of the LD group, MD group, and HD group were significantly increased, and there was a significant difference. CTX significantly decreased the microbial diversity in the intestinal tract of mice, which may have formed a relatively stable colony with low diversity. However, after continuous administration of intervention substances, the intestinal microbial diversity of mice in the intervention group increased, which may be related to the inhibiting the growth of some bacteria and promoting the growth of others, thus increasing their diversity.

Principal component analysis relies on the sample similarity coefficient to find principal coordinates, while principal coordinate analysis (PCoA) relies on the distance matrix to find principal coordinates. As shown in Figure 13B, the contribution value of the first major factor of distance PCoA to the sample difference in this study was 19.77%, and the contribution value of the second major factor to the sample difference was 12.21%. Results: The NC group and CTX group had obvious separation, indicating that the species difference between the two groups was obvious. After the administration of MLDs, the patients in the MD group were close to the NC group, and the MD group was the closest to the NC group, which had the effect of improving the intestinal flora.

The composition of intestinal flora in mice was analyzed, and the results are shown in Figure 14. At the phylum level, the ratio of firmicutes to Bacteroidetes in the CTX group was significantly higher than that in the NC group, while the thick-wall/pseudo-rods in the model group decreased after MLD treatment. At the family level, the relative abundance of Muribaculaceae and Staphylococcaceae increased significantly in the CTX group compared with the NC group. After MLD treatment, the relative abundance of Muribaculaceae and Staphylococcaceae was significantly lower than that of the CTX group. Compared with the NC group, the relative abundance of Bacteroidaceae in the CTX group was significantly reduced. After MLD treatment, the relative abundance of Bacteroidaceae was significantly higher than that of the CTX group. Sex differences will lead to differences in immune function, and the specific flora is a large part of the explanation. Fransen et al. [51] transferred the intestinal microbiota of male and female mice in a normal state to germ-free (GF) animals of the same or opposite sex. It was confirmed that sex differences in the microbiota caused sex-specific intestinal microbiota composition.

## 4. Conclusions

The specific microstructure of melanoidins has not been described yet. The extraction process of black garlic melanoidins was optimized. After preparation, three fractions were separated by DA201-C macroporous resin, and the molecular weight of the extracted MLDs was greater than 7 kDa. The results of the infrared analysis showed that the whole nigra was a mixture with a similar structure, and the infrared maps of different groups were similar, but the difference was obvious in the area below 1050 cm^−1^. In addition, their microscopic morphologies are curled, irregularly fragmented, and large sheets, respectively.

The strong physiological protection function of the immune system is the guarantee of the body’s health, which enables it to avoid the invasion of pathogens to a large extent. In this study, MLDs purified from black garlic were used to test their regulatory effect on immunocompromised mice, which confirmed the immune efficacy of MLDS. Black garlic MLDs can restore the proliferation and phagocytosis of macrophages, give lymphocytes higher activity, alleviate the disorder of serum factor (IFN-γ, IL-10, and TNF-α) levels caused by CTX, regulate the structure of intestinal flora, and improve the unhealthy level of intestinal SCFAs so that the immune system as a whole reaches a better state. In addition, we found that there were differences in the regulatory effects after ingestion of MLDs due to different mouse genders. Compared with female mice, MLDs have a more significant effect on the immune regulation of immunodeficient male mice, and more detailed studies are required to clarify the reasons for this. At present, some studies have shown that certain natural products have a very considerable effect on the treatment of diseases. Moreover, natural products have low toxicity and side effects, so they have a good application prospect. The immunomodulatory effect of MLDs in this study provides new insights into the exploitation and utilization of natural products.

## Figures and Tables

**Figure 1 foods-12-02004-f001:**
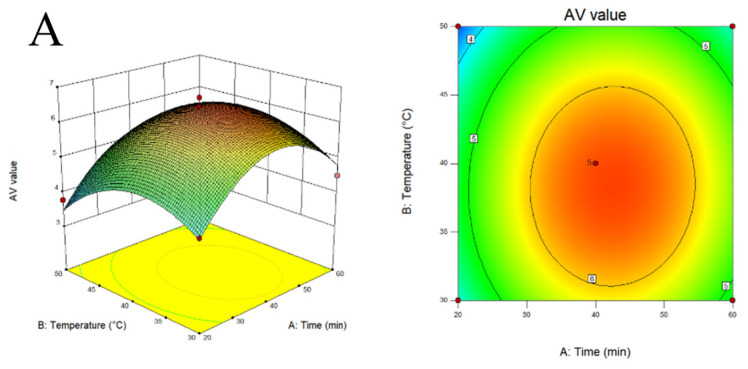
The response surface plots and contour plots. (**A**) Time and temperature, (**B**) time and solid–liquid ratio, (**C**) solution concentration and time, (**D**) temperature and solid–liquid ratio, (**E**) temperature and solution concentration, (**F**) solution concentration and solid–liquid ratio.

**Figure 2 foods-12-02004-f002:**
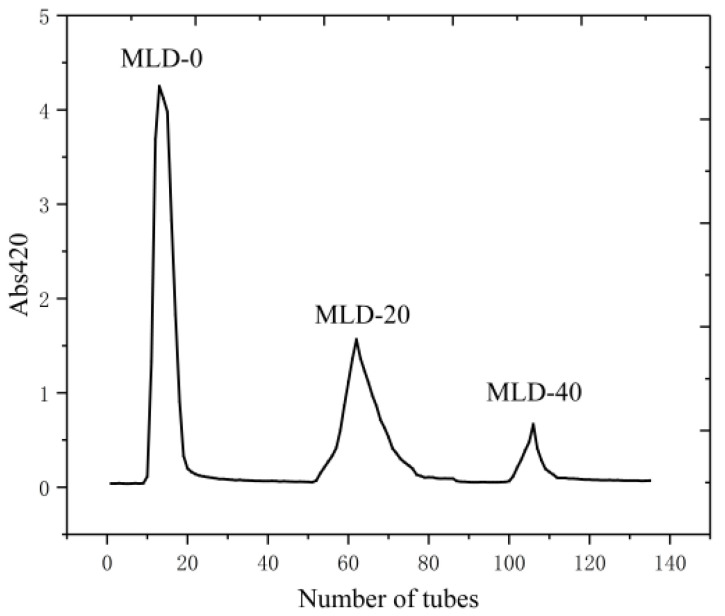
Chromatography of the black garlic melanoidins by DA201-C macroporous resin.

**Figure 3 foods-12-02004-f003:**
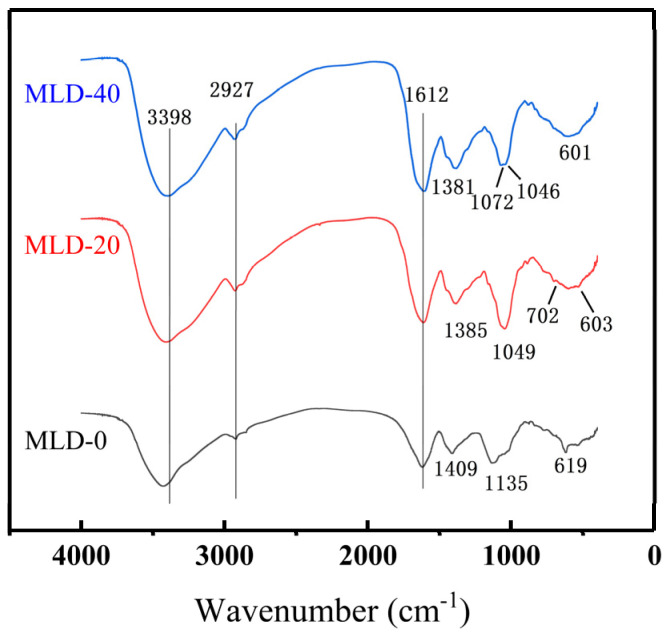
FTIR spectra of different melanoidin fractions.

**Figure 4 foods-12-02004-f004:**
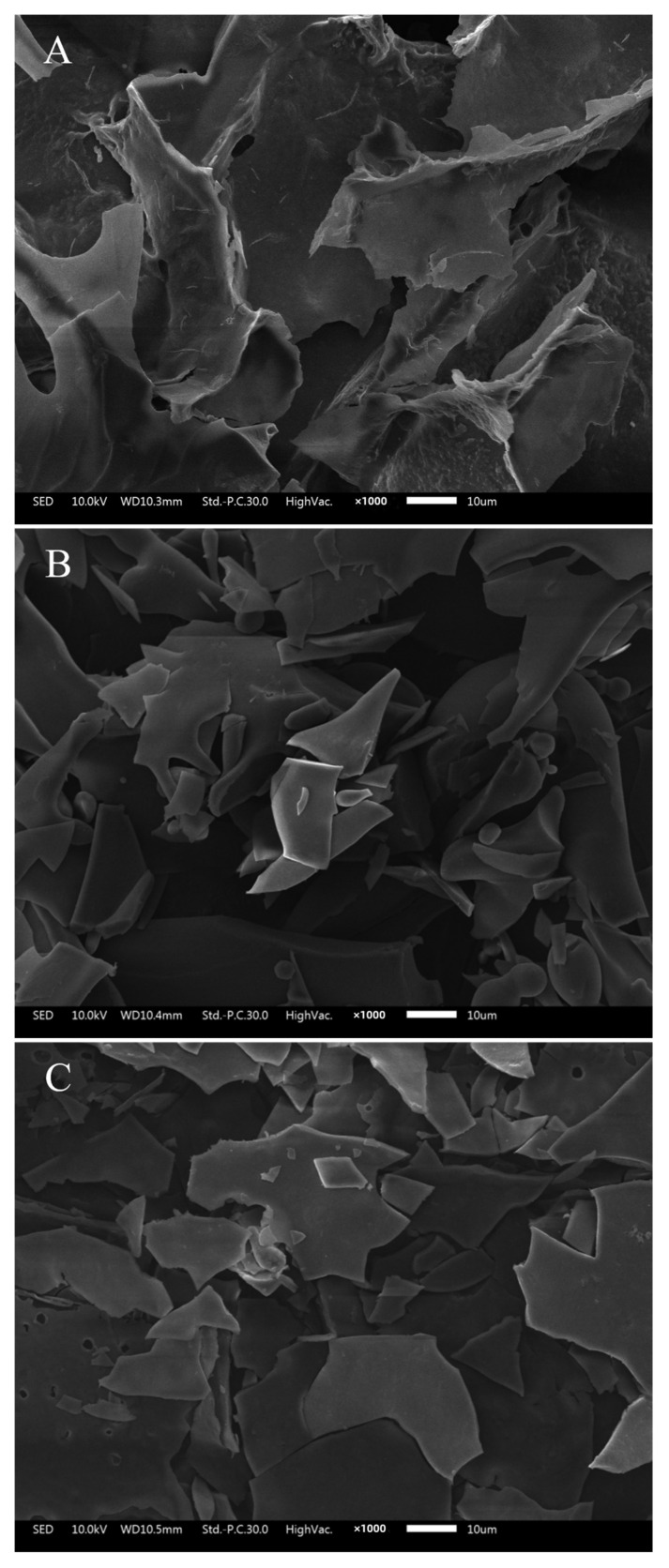
FTIR spectra of different melanoidin fractions. (×1000 magnification). (**A**) MLD-0, (**B**) MLD-20, and (**C**) MLD-40.

**Figure 5 foods-12-02004-f005:**
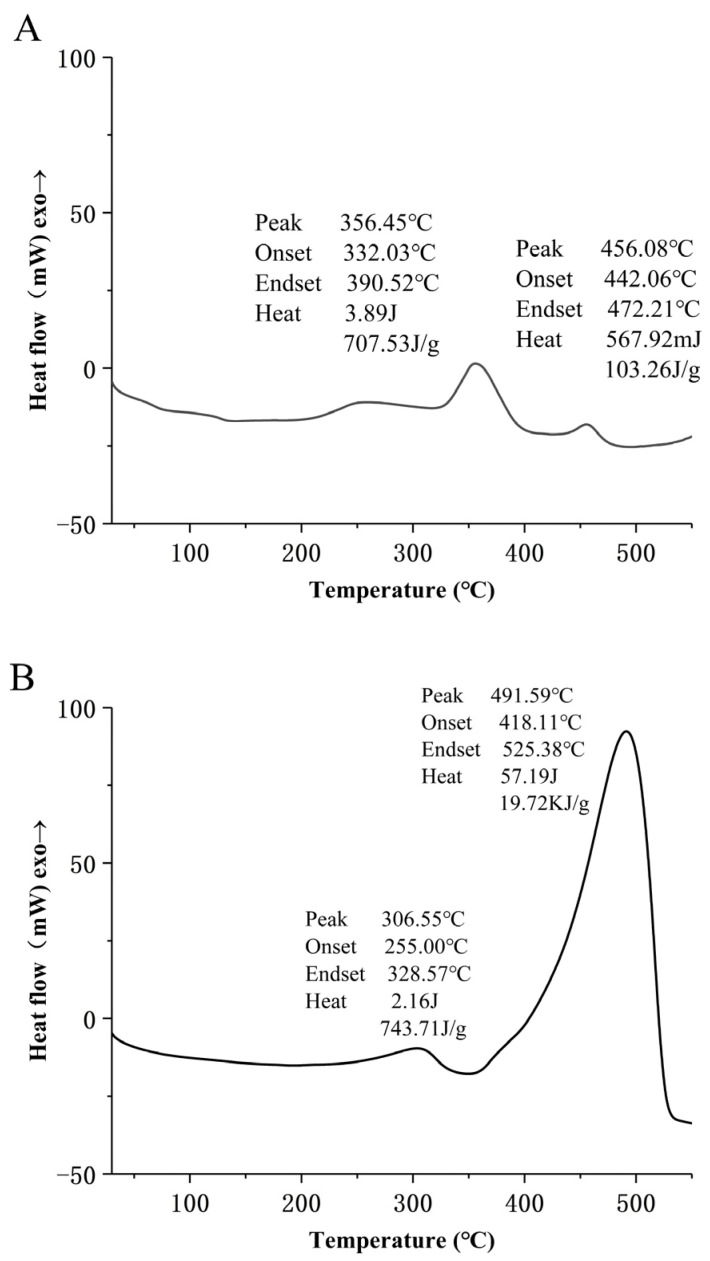
DSC curve of different melanoidin fractions. (**A**) MLD-0; (**B**) MLD-20; (**C**) MLD-40.

**Figure 6 foods-12-02004-f006:**
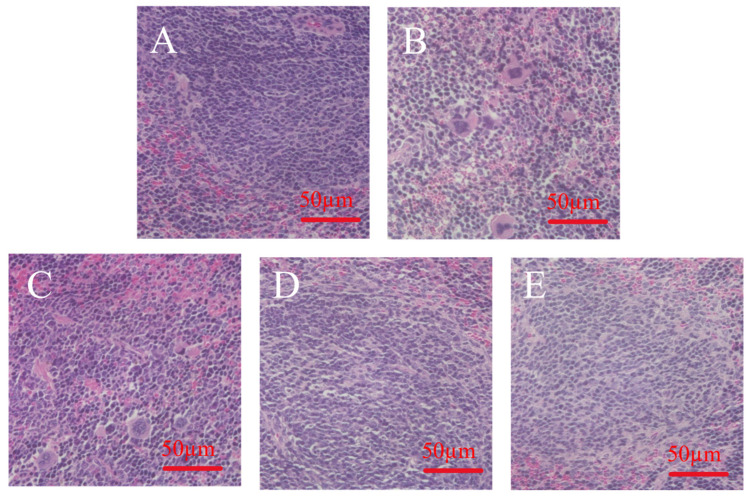
H&E staining of the spleen. (**A**) NC group; (**B**) CTX group, (**C**) LD group, (**D**) MD group; (**E**) HD group.

**Figure 7 foods-12-02004-f007:**
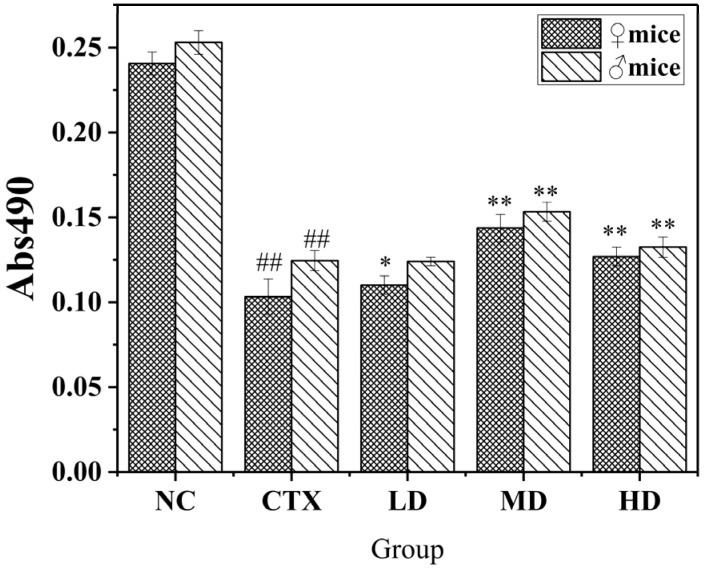
Effects of MLDs on macrophage proliferation in CTX-induced immunosuppressed mice. NC: normal group, CTX: model group, LD: low-dose MLD group, MD: medium-dose MLD group, HD: high-dose MLD group. Data are presented as mean ± SD, *n* = 12, ^##^
*p* < 0.01 vs. normal group; * *p* < 0.05, ** *p* < 0.01 vs. CTX group.

**Figure 8 foods-12-02004-f008:**
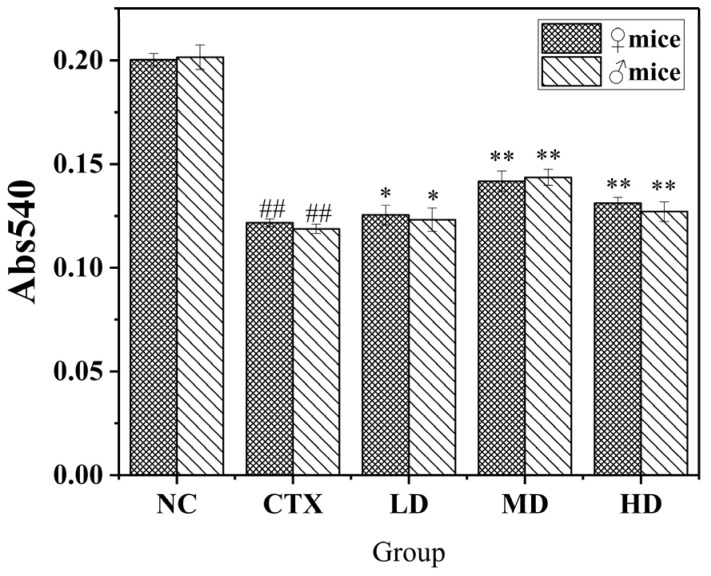
Effects of MLD on macrophage phagocytosis in CTX-induced immunosuppressed mice. NC: normal group, CTX: model group, LD: low-dose MLD group, MD: medium-dose MLD group, HD: high-dose MLD group. Data are presented as mean ± SD, *n* = 12, ^##^
*p* < 0.01 vs. normal group; * *p* < 0.05, ** *p* < 0.01 vs. CTX group.

**Figure 9 foods-12-02004-f009:**
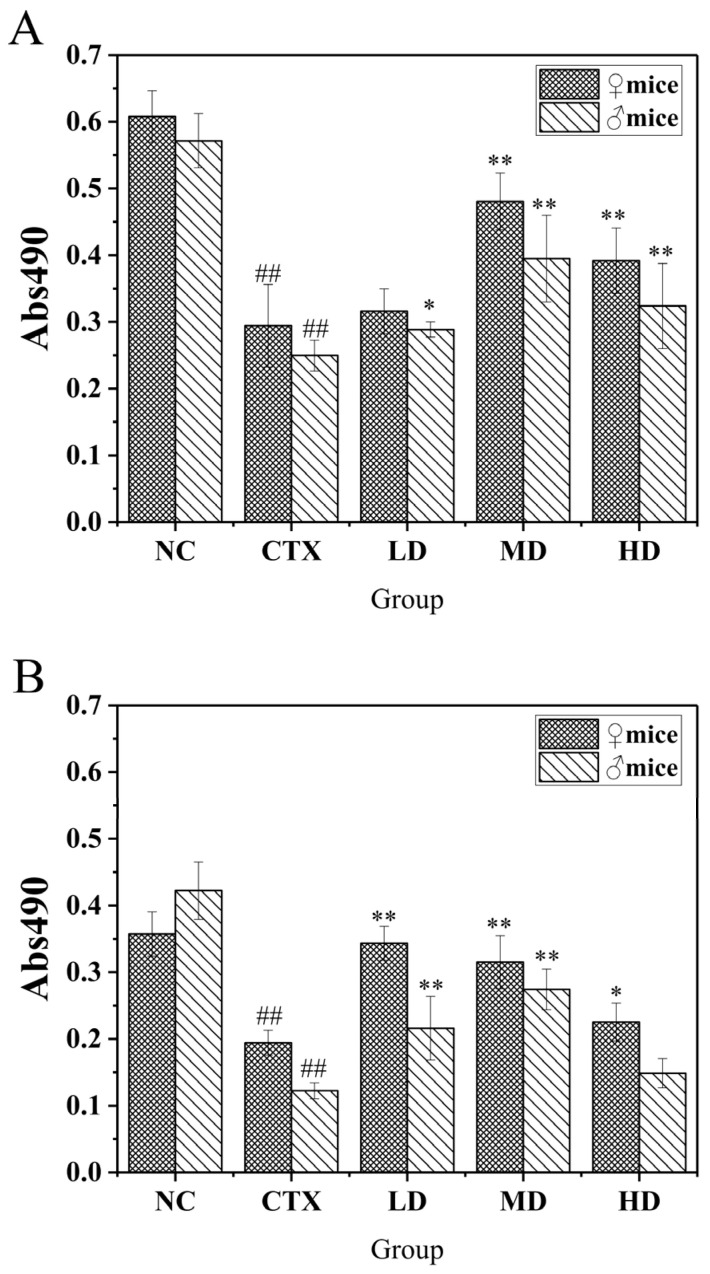
Effects of MLDs on the proliferative activity of splenic lymphocytes in CTX-induced immunosuppressed mice. (**A**,**B**) cell; B:T cell. NC: normal group, CTX: model group, LD: low-dose MLD group, MD: medium-dose MLD group, HD: high-dose MLD group. Data are presented as mean ± SD, *n* = 12, ^##^
*p* < 0.01 vs. normal group; * *p* < 0.05, ** *p* < 0.01 vs. CTX group.

**Figure 10 foods-12-02004-f010:**
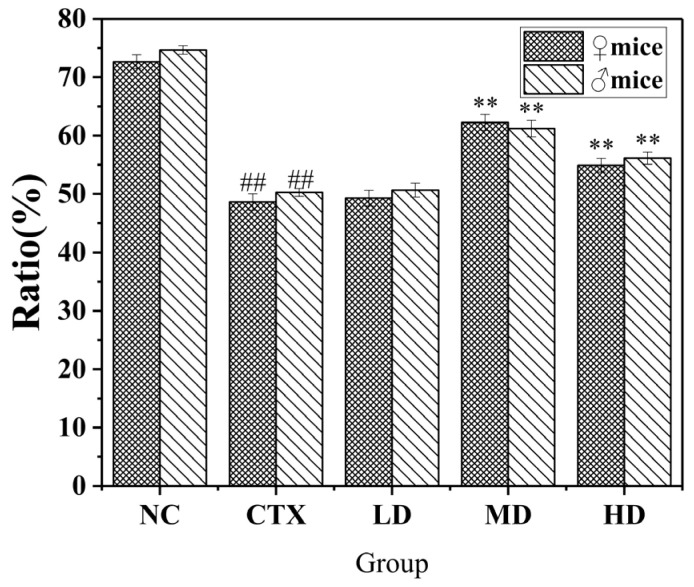
Effects of MLDs on NK cytotoxicity in CTX-induced immunosuppressed mice. NC: normal group, CTX: model group, LD: low-dose MLD group, MD: medium-dose MLD group, HD: high-dose MLD group. Data are presented as mean ± SD, *n* = 12, ^##^
*p* < 0.01 vs. normal group; ** *p* < 0.01 vs. CTX group.

**Figure 11 foods-12-02004-f011:**
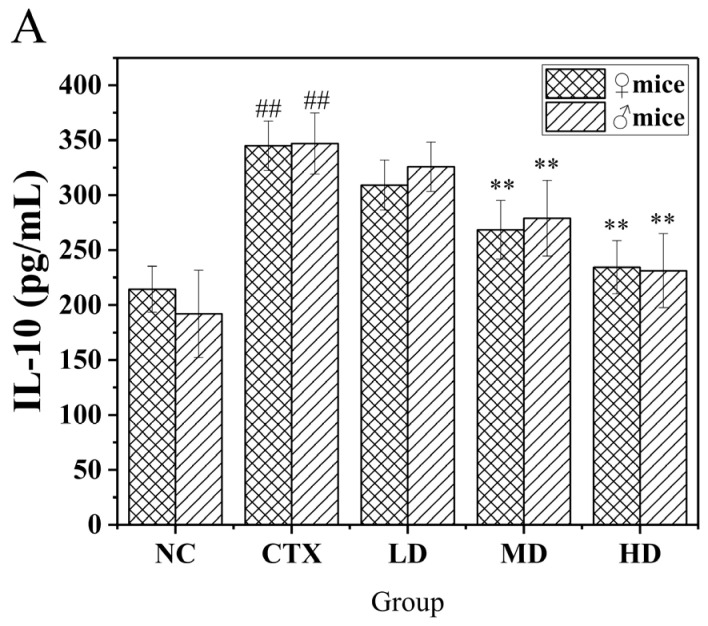
Effects of MLDs on the secretion of cytokines in CTX-induced immunosuppressed mice. (**A**) IL-10; (**B**) TNF-α; (**C**) IFN-γ. NC: normal group, CTX: model group, LD: low-dose MLD group, MD: medium-dose MLD group, HD: high-dose MLD group; ^##^
*p* < 0.01 vs. normal group; * *p* < 0.05, ** *p* < 0.01 vs. CTX group.

**Figure 12 foods-12-02004-f012:**
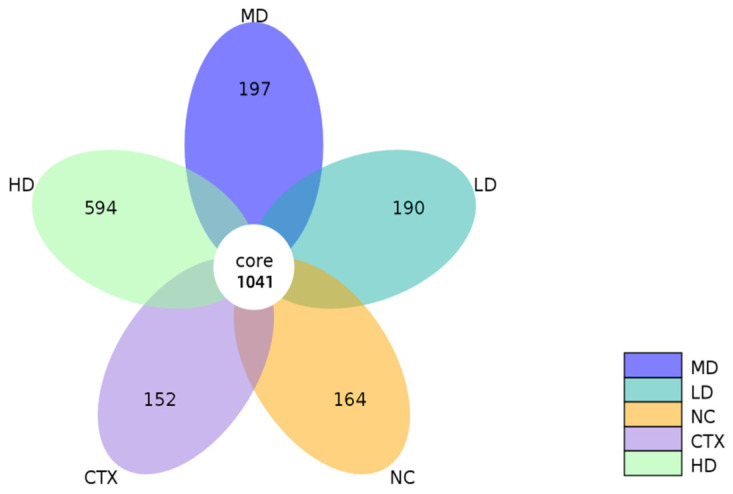
OTUs analysis of intestinal microbiota. NC: normal group, CTX: model group, LD: low-dose MLD group, MD: medium-dose MLD group, HD: high-dose MLD group.

**Figure 13 foods-12-02004-f013:**
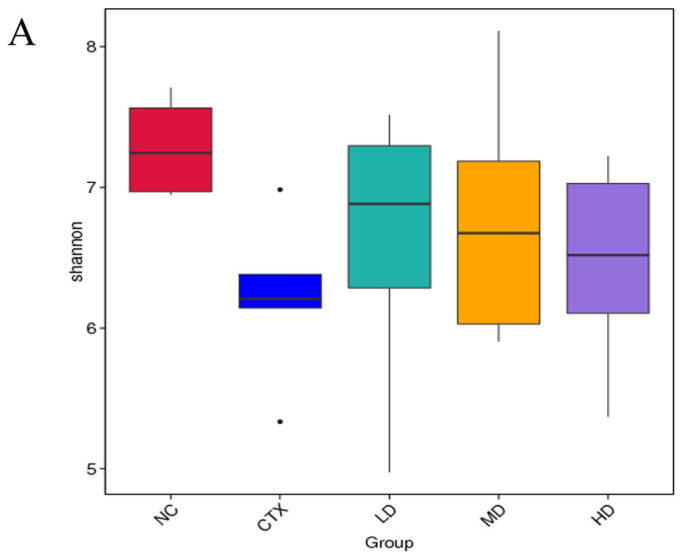
(**A**) Box plot of Shannon index differences between groups; (**B**) PCoA analysis of intestinal flora. NC: normal group, CTX: model group, LD: low-dose MLD group, MD: medium-dose MLD group, HD: high-dose MLD group.

**Figure 14 foods-12-02004-f014:**
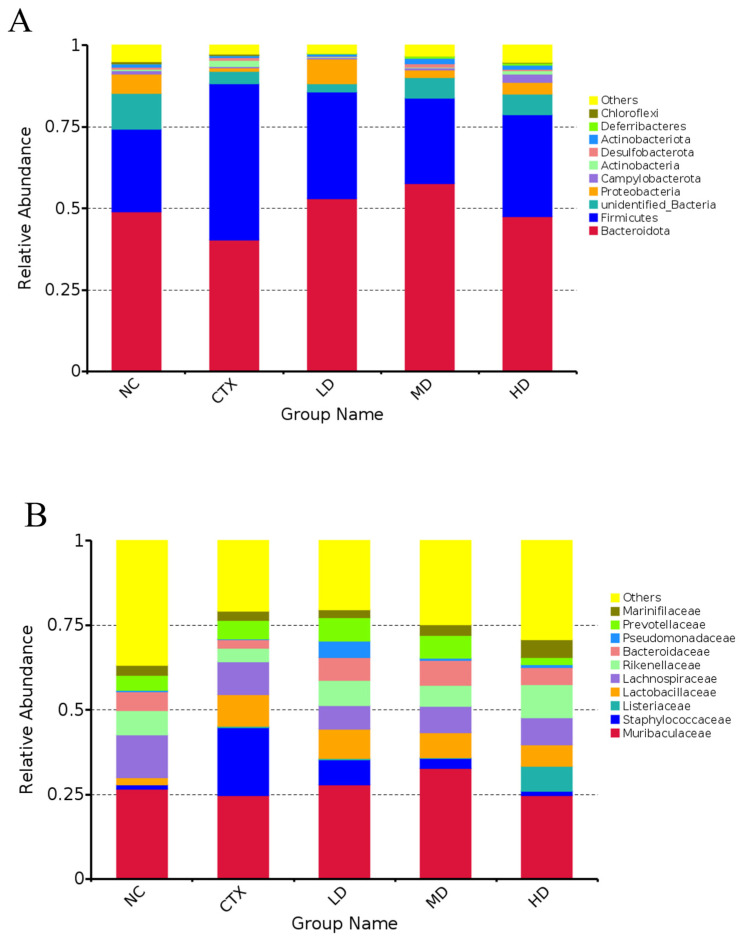
Effect of MLDs on gut microbial composition in immunosuppressed mice. (**A**) At the phylum level; (**B**) at the family level. NC: normal group, CTX: model group, LD: low-dose MLD group, MD: medium-dose MLD group, HD: high-dose MLD group.

**Table 1 foods-12-02004-t001:** Design of Box–Benhnken experiment.

Level	Primer			
	Time (min)	Temperature (°C)	Solid–Liquid Ratio	Ethanol Solution Concentration (%)
−1	20	30	1:22	10
0	40	40	1:18	20
1	60	50	1:14	30

**Table 2 foods-12-02004-t002:** The molecular weight of different MLD fractions.

Group	MLD	MLD-0	MLD-20	MLD-40
X = retention time (rt)	17.024	17.149	17.099	17.144
Y = LgMW	3.899898	3.856245	3.873706	3.857991
MW (kDa)	7.9	7.2	7.5	7.2

**Table 3 foods-12-02004-t003:** Effects of MLDs on the body weight and immune organ index in CTX-induced immunosuppressed mice (*n* = 8).

Gender	Group	Initial Body Weigh (g)	Final Body Weight (g)	Index of Spleen (mg/g BW)	Index of Thymus (mg/g BW)
Female mouse (♀)	NC	19.33 ± 0.93	20.29 ± 1.56	11.12 ± 0.93	3 ± 0.87
	CTX	19.27 ± 0.58	17.79 ± 1.38 ^##^	7.17 ± 0.62 ^##^	2.25 ± 0.62
	LD	19.21 ± 1.1	19.08 ± 1.11 *	8.4 ± 0.91	2.62 ± 1.48
	MD	19.27 ± 1.11	19.33 ± 0.68 *	9.28 ± 1.62 **	2.87 ± 1.07
	HD	19.3 ± 0.88	18.67 ± 0.98	9.89 ± 1.21 **	2.72 ± 0.48
Male mouse (♂)	NC	23.7 ± 0.66	24.28 ± 1.38	9.2 ± 1.58	1.51 ± 0.42
	CTX	24.01 ± 1.24	20.85 ± 1.19 ^##^	5.81 ± 0.34 ^##^	1 ± 0.62
	LD	23.94 ± 1.18	22.79 ± 1.41 **	6.83 ± 0.7	1.46 ± 0.25
	MD	23.89 ± 1.31	23.96 ± 1.03 **	7.47 ± 1.39 *	1.36 ± 0.38
	HD	23.36 ± 0.82	21.77 ± 1.14	7.71 ± 0.91 **	1.13 ± 0.46

Note: Data are expressed as mean ± SD, *n* = 8, ^##^
*p* < 0.01 vs. normal group; * *p* < 0.05, ** *p* < 0.01 vs. CTX group. NC: normal group, CTX: model group, LD: low-dose MLD group, MD: medium-dose MLD group, HD: high-dose MLD group.

**Table 4 foods-12-02004-t004:** Effects of MLDs on SCFAs in CTX-induced immunosuppressed mice.

ug/g	NC	CTX	LD	MD	HD
Acetic Acid	5.09 ± 0.04	4.1 ± 0.13	3.42 ± 0.2	3.64 ± 0.67	6.15 ± 1.37 *
Propanoic Acid	8.12 ± 0.56	1.63 ± 0.1 ^##^	2.14 ± 0.1 *	2.92 ± 0.16 **	5.45 ± 0.13 **
Butyric Acid	2.03 ± 0.35	0.51 ± 0.04 ^##^	0.72 ± 0.09	0.68 ± 0.06	1.66 ± 0.21 ^##^
Valeric Acid	6.11 ± 0.47	0.75 ± 0.12 ^##^	2.57 ± 0.16 **	3.03 ± 0.3 **	4.38 ± 0.89 **

Note: NC: normal group, CTX: model group, LD: low-dose MLD group, MD: medium-dose MLD group, HD: high-dose MLD group. Data are expressed as mean ± SD, *n* = 6, ^##^
*p* < 0.01 vs. normal group; * *p* < 0.05, ** *p* < 0.01 vs. CTX group.

## Data Availability

Data is contained within the article.

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
