# Peer review of "Structural Characteristics and Immunomodulatory Effects of Melanoidins from Black Garlic"

_foods, 2023, doi:10.3390/foods12102004_

Round 1

Reviewer 1 Report

Comments to authors

By reviewing the paper entitled:

" Structural characteristics and immunomodulatory effects of melanoidins from black garlic"

The following remarks have been recorded:

Abstract:

-         Page 1, Line 24: please, add a clear statement projecting the conclusion.

Keywords, Page 1, Line 26: please, replace "enhance immunity" with "Immunity enhancers"

Introduction:

-         Page 2, Lines 53-56: Please, reform the paragraph ( The growth state and organ …………………melanoidins on immune ability) to present the aim of the study in a simple and clear way, for instance, The aim of the current study is.......................... 

 MM section:

-          Please, add the details of the device (s) used and the details of the manufacturer of devices or chemicals used (throughout the entire MM section).

-         Delete (Beijing).

-         Page 3, Lines 118-119: replace "light and darkness are alternated for 12 hours" with "12 hours Light/ dark cycle"

-          Page 4, Line 131: replace "death" by "sacrificing"

-         Page 4, Line 134: replace "complete" by "whole"

-         Page 4, Lines 139-143: Please, add a reference for histological techniques

-         Page 4, Lines 143: Since the photomicrographs included in the present paper were stained with HE, I think the microscope applied is ordinary compound light microscope rather than fluorescent microscope.

-          Page 4, Lines 146-150: Please, remove this paragraph from MM section. This should be in the introduction and/or the discussion.

-         Page 4, Lines 168: replace "The activity of mouse spleen cells was determined" by " Determination of the activity of mouse spleen lymphocytes"

-         Page 5, Lines 186-190: Please, remove this paragraph from MM section. This should be in the introduction and/or the discussion.

-         Page 5, Lines 216-221: Please, remove this paragraph from MM section. This should be in the introduction and/or the discussion.

-         Page 6, Lines 237-244: Please, remove this paragraph from MM section. This should be in the introduction and/or the discussion.

-         Page 6, Lines 253-257: Please, add the details of the version of software applied and all other details of firms or owners 

Results:

-         Fig 2, Pages 7-8: Is this Fig. 1 or 2? if it's fig 2, where is Fig. 1. Otherwise, renumber the figures in legends and where they have been mentioned within the text. Please, collect all images to be in one panel. Also, the labeling font is too small to read. Please modify it to be readable

-         Fig 4, page 9: the labeling font in the figure is too small to read. Please modify it to be readable

-         Fig 5, page 10: the labeling font in the figure is too small to read. Please modify it to be readable, especially the scale bars.

-         Fig 6, page 10: the labeling font in the figure is too small to read. Please modify it to be readable.

-         Table 3, page 11, line 358-361: replace "of immunosuppressive mice induced by CTX" by "in CTX-induced immunosuppressed mice"

-         Table 3, page 11, line 358-361: I suggest to collect both male and female in one table as follows by adding a column of "sex" or "gender" next to the column of Groups.

-         Fig 7, page 12: Labeling is missed. Also, the scale bars are not clear

-         Fig 8, page 12: the labeling font of the histogram is too small to read. Please modify it to be readable

-         Fig 8, page 12, line 396: replace "of immunosuppressive mice induced by CTX" by "in CTX-induced immunosuppressed mice"

-         Fig 9, page 13: the labeling font of the histogram is too small to read. Please modify it to be readable

-         Fig 9, page 13, line 413: replace "of immunosuppressive mice induced by CTX" by "in CTX-induced immunosuppressed mice"

-         Fig 10, page 14: the labeling font of the histogram is too small to read. Please modify it to be readable

-         Fig 10, page 14, line 435-436: replace "in immunosuppressive mice induced by CTX" by "in CTX-induced immunosuppressed mice"

-         Fig 11, page 14: the labeling font of the histogram is too small to read. Please modify it to be readable

-         Fig 11, page 14, line 451: replace " immunosuppressive mice induced by CTX" by " CTX-induced immunosuppressed mice"

-         Fig 12, page 15: the labeling font of the histogram is too small to read. Please modify it to be readable

-         Fig 12, page 15, line 464: replace " immunosuppressive mice induced by CTX" by " CTX-induced immunosuppressed mice"

-         Page 15, Lines 468-471: Please, remove this paragraph. It should be in the "Introduction" or "Discussion" section.

-         Page 15, Line 486: it should be table 4 not table 3, the numbering of tables should be carefully reviewed. Also, replace "of immunosuppressive mice induced by CTX" by "in CTX-induced immunosuppressed mice"

-          Pages 15-16, Lines 489-504: Please, remove this paragraph. It should be in the "Introduction" or "Discussion" section.

-         Page 17, Line 528: Please, replace "Hannon" with "Shannon". Also, the labeling font of the histogram is too small to read. Please modify it to be readable

-         Fig 15, Page 17: the labeling font of the histogram is too small to read. Please modify it to be readable.

Discussion:

-          Page 18, Lines 555-569: The Discussion is too short and has not been provided with the citations, which support or contradict the present findings. Discussion should be rewritten

Conclusion:

-         Page 18, Lines 570: Where is the conclusion? Please, provide a paragraph projecting the conclusion and the recommendation of your study.

 For Additional modification remarks, please follow the comments within the attached pdf version of the paper.

For the abovementioned remarks, I recommend a major revision.

Reviewer 2 Report

Dear authors,

The current work still needs to concern the plagiarism based on the PDF evident attachment file herewith. Please check it out in the whole manuscript, especially the methods section. Also, some point need to make the clarify by the authors are....

1. The figures should increase the resolution and the bigger letter or number

2. What is the reason behind the H&E determination of the spleen of the animal??., it seems the results as presented in the current work were not different.? please justify.

3. In general, when we consume black garlic, the main chemical or bioactive component is not melanoidin's.? so how does it affect on the immune system of mammalians?

4. The discussion of this current work seems too very short and some significant points of the study did not mention or highlighted. May the author needs to discuss the vital data based on the finding.

5. Please check the discussion and conclusion of the study. may some error be detected? the conclusion was not included in the paper?

Reviewer 3 Report

Abstract starts with "In this experiment....". Should be revised. 

M&M section needs to be thoroughly revised, excluding the unnecessary discussion in the beginning of many sub-sections. 

Figure legends lack detailed explanation. 

Figures are very hard to understand since font size is, very often, extremely small. 

Section 3.10 Effect of MLD on spleen cells in mice, is completely lacking text. 

Discussion is completely unadequate.

Conclusions are lacking. 

Round 2

Reviewer 1 Report

Thank you for addressing most of the comments. However, some minor issues should also be done.

1) In MM section Page 5, Line 198: please, add a reference for histological techniques.

2) In Results & Discussion section: Page 15, Line 470: Please delete " x30 magnification ", the scale bars on images are sufficient.

3) In Results & Discussion section: Page 24, Fig 14:  the labeling font in the figure is too small to read. Please modify it to be readable

.

Author Response

Thanik you for the comments concerning our manuscript entitled“Structural characteristics and immunomodulatory effects of melanoidins from black garlic”(foods-2332905).Those comments are all valuable and very helpful for revising and improving our papeI, as well as the important guiding significance to our researches.We have studied comments carefully and have made correctionis which we hope meet with approval. Revised portions are marked in yellow on the paper. The corrections in the paper and the responses to the reviewer's comments are as flowing:

  1. In MM section Page 5, Line 198: please, add a reference for histological techniques.

Response:Thank you for this comment. References to histological techniques have been added to the article.(Page 5, Line 198)

Here are the details: The proliferation capacity of lymphocytes was determined by subtracting the ab-sorbance values of the blank control group from the measured absorbance values of the T lymphocyte group or the B lymphocyte group [26].

  1. In Results & Discussion section: Page 15, Line 470: Please delete " x30 magnification ", the scale bars on images are sufficient.

Response:Thank you for this comment. " x30 magnification " has been deleted accordingly in the article.(Page 15, Line 382)

  1. In Results & Discussion section: Page 24, Fig 14:  the labeling font in the figure is too small to read. Please modify it to be readable.  

 Response:Thank you for this comment. TThe image size has been changed in this article to make the tag font readable.(Page 22, Line 591-592)

Reviewer 3 Report

Dear authors,

The figure legends are still inadequate. The legends should support figures as free-standing and understandable without reading any part of the manuscript. 

Author Response

Thanik you for the comments concerning our manuscript entitled“Structural characteristics and immunomodulatory effects of melanoidins from black garlic”(foods-2332905).The comments are all valuable and very helpful for revising and improving our papeI, as well as the important guiding significance to our researches.We have studied comments carefully and have made correctionis which we hope meet with approval. Revised portions are marked in yellow on the paper. The corrections in the paper and the responses to the reviewer's comments are as flowing:

  1. The figure legends are still inadequate. The legends should support figures as free-standing and understandable without reading any part of the manuscript. 

Response:Thank you for this comment. We have made more clear notes on the article ICONS respectively in: Table 3 (12 pages, 364 lines), Table 4 (19 pages, 511 lines), Figure 7 (14 pages, 405 lines), Figure 8 (15 pages, 423 lines), Figure 9 (16 pages, 488 lines), Figure 10 (17 pages, 469 lines), Figure 11 (18 pages, 485 lines), Figure 12 (20 pages, 550 lines), Figure 13 (21 pages, 566 lines), Figure 14 (page 22, line 594).